# Effect of Phenylalanine–Arginine Beta-Naphthylamide on the Values of Minimum Inhibitory Concentration of Quinolones and Aminoglycosides in Clinical Isolates of *Acinetobacter baumannii*

**DOI:** 10.3390/antibiotics12061071

**Published:** 2023-06-18

**Authors:** Stefany Plasencia-Rebata, Saul Levy-Blitchtein, Juana del Valle-Mendoza, Wilmer Silva-Caso, Isaac Peña-Tuesta, William Vicente Taboada, Fernando Barreda Bolaños, Miguel Angel Aguilar-Luis

**Affiliations:** 1Escuela de Medicina, Facultad de Ciencias de la Salud, Universidad Peruana de Ciencias Aplicadas, Lima 15023, Peru; plasenciarebata@gmail.com (S.P.-R.); levysaul45@hotmail.com (S.L.-B.); wilmer.silva@upc.pe (W.S.-C.); 2Microbiology Department, Vall d’Hebron University Hospital, 08034 Barcelona, Spain; 3Laboratorio de Biomedicina, Facultad de Ciencias de la Salud, Universidad Peruana de Ciencias Aplicadas, Lima 15023, Peru; isaacp1503@gmail.com; 4Instituto de Investigación Nutricional, Lima 15024, Peru; 5National Institute of Neoplastic Diseases, Lima 15024, Peru; willyvita@hotmail.com (W.V.T.); fernando.barreda73@hotmail.com (F.B.B.)

**Keywords:** *Acinetobacter baumannii*, efflux pump inhibitors, antimicrobial resistance, multidrug resistance

## Abstract

(1) Background: *Acinetobacter baumannii* has become the most important pathogen responsible for nosocomial infections in health systems. It expresses several resistance mechanisms, including the production of β-lactamases, changes in the cell membrane, and the expression of efflux pumps. (2) Methods: *A. baumannii* was detected by PCR amplification of the blaOXA-51-like gene. Antimicrobial susceptibility to fluoroquinolones and aminoglycosides was assessed using the broth microdilution technique according to 2018 CLSI guidelines. Efflux pump system activity was assessed by the addition of a phenylalanine–arginine beta-naphthylamide (PAβN) inhibitor. (3) Results: A total of nineteen *A. baumannii* clinical isolates were included in the study. In an overall analysis, in the presence of PAβN, amikacin susceptibility rates changed from 84.2% to 100%; regarding tobramycin, they changed from 68.4% to 84.2%; for nalidixic acid, they changed from 73.7% to 79.0%; as per ciprofloxacin, they changed from 68.4% to 73.7%; and, for levofloxacin, they stayed as 79.0% in both groups. (4) Conclusions: The addition of PAβN demonstrated a decrease in the rates of resistance to antimicrobials from the family of quinolones and aminoglycosides. Efflux pumps play an important role in the emergence of multidrug-resistant *A. baumannii* strains, and their inhibition may be useful as adjunctive therapy against this pathogen.

## 1. Introduction

*Acinetobacter* spp. are gram-negative coccobacilli, non-fermenters, non-motile, oxidase-negative, and catalase-positive and are classified under the Moraxellaceae family [1]. *A. baumannii*, *Acinetobacter calcoaceticus*, *A. nosocomialis*, *A. pittii*, *A. seifertii*, and *Acinetobacter dijkshoorniae* comprise 90% to 95% of clinically significant infections [2,3]. 

*A. baumannii* is an emerging nosocomial pathogen that has developed mechanisms to resist disinfection, desiccation, and oxidative stress [4]. Its ability to survive under a wide range of environmental conditions and to persist for extended periods of time on surfaces makes it a frequent cause of outbreaks and an endemic healthcare-associated pathogen [5], whose clinical significance has been increasing in the last three decades [6]. Institutional outbreaks caused by multidrug-resistant (MDR) strains are a growing public health concern [1]. This bacterium has currently become the most important pathogen responsible for nosocomial infections in the health system worldwide [7].

*A. baumannii* causes ventilator-associated pneumonia or central-line bloodstream infections and, less frequently, skin and soft tissues infections, such as war wound infections, surgical site infections, and catheter-associated urinary tract infections [5,7,8,9,10,11]. Risk factors regarding colonization or infection with multidrug-resistant species include prolonged hospitalization, admission to the intensive care unit (ICU), mechanical ventilation, long-term exposure to broad-spectrum antibiotics, recent surgery, invasive procedures, and underlying severe illnesses [1,5,8,9,10,11,12,13].

Several intrinsic and acquired resistance mechanisms are frequently expressed in nosocomial strains. These mechanisms include an increased production of antibiotic efflux pumps [14], point mutations in target proteins to inactivate antibiotic effects, enzymatic modification, antimicrobial degradation, and a reduction in membrane permeability [15,16]. In order to cause clinical resistance in Acinetobacter, efflux pumps usually act in association with the overexpression of Amp C β-lactamases or carbapenemases. In addition to removing β-lactam antibiotics, efflux pumps can actively expel macrolides, quinolones, tetracyclines, chloramphenicol, and disinfectants [13].

Regarding clinical resistance, it has been reported that the rates of resistance to multiple drugs in *A. baumannii* are approximately four times higher than in other gram-negative bacilli [17]. Added to this is a resistance to carbapenem antibiotics, whose accelerated diffusion not only in Latin America but worldwide represents a serious clinical threat [18]. It is considered that resistance against carbapenem antibiotics is, in itself, sufficient to define an isolate of *A. baumannii* as highly resistant and therefore it is important to describe its presence. The presence of resistance in this microorganism is multifactorial and is largely due to the production of carbapenemases (metallo-β-lactamases, oxacillinases) [19]. More than six types of metallo-β-lactamases (MBL) have been described in *A. baumannii* around the world [16]. Regarding oxacillinases, five types have been described: OXA-23, OXA-24/40, OXA-58, OXA-143, and OXA-235, all capable of hydrolyzing carbapenems [19]. Unlike enzymatic mechanisms of resistance, non-enzymatic mechanisms, such as efflux pumps, an alteration or change in penicillin-binding protein (PUP), and the loss of outer membrane protein (OMP), are also responsible for bacterial resistance to drugs and are little analyzed or not analyzed in clinical microbiology laboratories. All these factors contribute to the failure of antibiotic treatment [20,21].

In this context, the most realistic approach in recent years is to investigate resistance inhibition rather than to synthesize new compounds. In this category, efflux pumps stand out [22]. Efflux pumps usually have three components and *A. baumannii* may contain four categories of efflux pumps, including the RND, MFS, MATE, and SMR family transporters, which have been reported to be related to *A. baumannii*’s antimicrobial resistance, which is capable of actively pumping out a broad range of antimicrobial and toxic compounds from the cell [23]. Five different families of transport proteins have been shown to include multidrug efflux systems [24]. Recently, the proteobacterial antimicrobial compound efflux (PACE) family has been described as the sixth family of bacterial multidrug efflux systems [24]. RND-type transporters are known to play a dominant role in the MDR of many *Acinetobacter* species [25]. While the overexpression of Ade transporters is often beneficial to bacteria, this is not always the case; some Ade transporters, such as AdeABC, AdeFGH, and AdeIJK, can be toxic to cells when overexpressed [26,27]. Therefore, to assess the role of the drug efflux mechanism in bacteria, efflux pump inhibitors (EPIs) are widely used [28]. The effects of several EPIs, including carbonyl cyanide m-chlorophenylhydrazone (CCCP), phenylalanine–arginine beta-naphthylamide (PAβN) [29], and 1-1-napthylmethyl)-piperazine (NMP) [30,31,32], have been previously assessed in a small number of in vitro studies, including CCCP, PaβN [29], and NMP, along with other drugs that may impact efflux mechanisms (omeprazole, verapamil, reserpine, phenothiazines) [30,31,32]. One of the best-studied EPIs is the peptidomimetic compound phenylalanine–arginine beta-naphthylamide (PAβN, also called MC207, 110), which was originally described in 1999 and was characterized in 2001 as a broad-spectrum efflux pump inhibitor that is capable of significantly reducing fluoroquinolone resistance [33] and permeabilizing membranes in *P. aeruginosa* [34].

Hence, the aim of this study was to analyze the contribution of the active efflux system to resistance to quinolones and aminoglycosides in selected epidemic *A. baumannii* clinical isolates with the efflux pump inhibitor PAβN.

## 2. Results

A total of nineteen *Acinetobacter baumannii* strains that were identified through the amplification of the blaOXA-51 gene were included.

The patterns of susceptibility and resistance according to each antibiotic in the absence and presence of the efflux pump inhibitor are described in Table 1.

The table describes the antimicrobial susceptibility to five antibiotics, three of them belonging to the quinolone family (nalidixic acid, ciprofloxacin, levofloxacin) and two aminoglycosides (tobramycin, amikacin). Two groups were formed for each of the nineteen strains, one group that evaluated the susceptibility in the presence of only the antibiotic and the other group that evaluated the susceptibility of the microorganism against the antibiotic plus the efflux pump inhibitor studied (PAβN).

Susceptibility was assessed according to the CLSI 2018 standards (Table 2) [35] for MIC values, as well as to the four-fold decrease in the minimum inhibitory concentration value. Based on CLSI 2018 guideline recommendations for disc diffusion, a maximum of 12 discs were used in a 150 mm plate and no more than 6 discs in a 100 mm plate; the discs were placed no less than 24 mm center to center apart. Each zone diameter was clearly measurable, and no overlapping zones were recorded that would prevent accurate measurement. The diameter of the complete inhibition zones was determined by eye, including the disc diameter. The Petri dish was held a few centimeters above a black illuminated background at the time of measurement. The margin of the zone was determined considering the area that does not show an evident and visible growth that can be detected with the naked eye. In the present study, the growth of tiny colonies was not recorded, but it was taken into account that, if the case occurs, the weak growth of tiny colonies should be ignored since they can only be detected with a magnifying glass at the edge of the zone of inhibited growth; therefore, the most evident margin was measured to determine the diameter of the zone.

Non-susceptibility rates in the absence of PAβN including intermediate (I) and resistant (R) categories were tobramycin (31.6%, n = 6), ciprofloxacin (31.6%, n = 6), levofloxacin (21.1%, n = 4), nalidixic acid (26.3%, n = 5), and amikacin (15.8%, n = 3).

In the first place, according to the 2018 CLSI, eight strains that were in the presence of PAβN showed a decrease in resistance to at least one antimicrobial evaluated. From this group, ciprofloxacin (21.1%, n = 4) was the drug that showed the highest frequency of an increase in susceptibility (Figure 1).

On the other hand, according to the four-fold decrease in the minimum inhibitory concentration value, a total of sixteen strains (84.2%) that were in the presence of the inhibitor showed decreased resistance profiles in at least one of the antimicrobials assessed, despite the fact that, in some cases, it did not imply that categories between sensitive, intermediate and resistant were changed. In this case, when only the MIC reduction is taken into consideration, more strains demonstrated changes in susceptibility to amikacin (36.8%, n = 7) and nalidixic acid (36.8%, n = 7), followed by levofloxacin (26.3%, n = 5), ciprofloxacin (21.1%, n = 4), and tobramycin (11.0%, n = 2). The inhibitory action of amikacin and nalidixic acid was potentiated by PAβN in seven strains, with a four-fold reduction in MIC. In one of the cases, the decrease in the MIC for three antibiotic agents simultaneously was described.

In an overall analysis, amikacin susceptibility rates changed from 84.2% (n = 16) to 100% (n = 19); as per tobramycin, they increased from 68.4% (n = 13) to 84.2% (n = 16), whereas the rates for nalidixic acid changed from 73.7% (n = 14) to 79.0% (n = 15), for ciprofloxacin, they changed from 68.4% (n = 13) to 73.7% (n = 14), and, for levofloxacin, they stayed as 79.0% (n = 15) in both groups.

Regarding antibiotics of the quinolone family, i.e., ciprofloxacin and levofloxacin, a paradoxical reduction in the death or inhibition of the bacteria due to antibiotics was observed in higher concentrations than their optimal bactericidal concentration. Three cases were recorded for ciprofloxacin and two cases for levofloxacin.

## 3. Discussion

Due to *Acinetobacter baumannii*, it is a worldwide concern to determine the mechanisms of resistance to antibiotics in the pathogenesis of this bacterium, and evaluating alternative treatment strategies against it has become a priority. The combined actions of outer membrane protein A, biofilm formation on biotic and abiotic surfaces, phospholipases C and D, the metal homeostatic system, lipopolysaccharides, and verotoxins have been determined to be relevant to the virulence and pathogenesis of this microorganism. Added to this is the fact that *A. baumannii* is resistant to broad-spectrum antibiotics due to its multiple mechanisms, such as β-lactamases, efflux pumps, aminoglycoside-modifying enzymes, permeability changes, and target alternation [36]. Thus, deepening the study of efflux pumps has led to the knowledge that they can be specialized for a single substrate or can transport a variety of structurally different molecules (including antibiotics of many classes) for which these pumps have been related to multidrug resistance (MDR) [37]. With this information, it can be established that the inhibition of the efflux pumps could increase the sensitivity to several classes of antibiotics and allow for a better therapeutic clinical response in patients with various infections caused by this microorganism [38]. Novel antimicrobial peptides, sterilization techniques, and combination therapies with efflux pumps are being developed to effectively treat *A. baumannii* infections [39]. In this context, the present study evaluated the sensitivity of a group of five antibiotics against *A. baumannii* in the presence and absence of an efflux pump inhibitor.

The results of various studies indicate that there are several types of multidrug efflux pumps. Among these, the efflux pumps belonging to the resistance–nodulation–cell division (RND) family stand out, which play an important role in the resistance to antibiotics of several species of Gram-negative bacteria [40,41]. Three different RND pumps are currently recognized in *A. baumannii*: AdeABC [42], AdeIJK [43], and, the third described, AdeFGH, in a mutant strain of this microorganism [44]. The AdeABC system has been shown to expel aminoglycosides such as amikacin, gentamicin, and kanamycin, as well as chloramphenicol, cefotaxime, erythromycin, and quinolones such as norfloxacin, ofloxacin, pefloxacin, and sparfloxacin, as well as netilmicin, tetracycline, tobramycin, and trimethoprim [42]. On the other hand, the AdeIJK pump expels beta-lactam drugs, fluoroquinolones, chloramphenicol, tetracycline, erythromycin, lincosamides, fusidic acid, novobiocin, rifampicin, trimethoprim, acridine, pyronin, safranin, and sodium dodecyl sulfate (SDS) [43]. Similarly, the AdeFGH bomb in a mutant strain of *A. baumannii* in the absence of the AdeABC and AdeIJK pumps was shown to expel chloramphenicol, trimethoprim, quinolones such as ciprofloxacin, and clindamycin [44,45]. On the other hand, the AdeDE pump reported in *Acinetobacter* group 3 confers resistance to amikacin, ceftazidime, chloramphenicol, ciprofloxacin, erythromycin, ethidium bromide, meropenem, rifampicin, and tetracycline [46].

Consistent with the findings of Peleg et al. [47] and Valentine et al. [48], ciprofloxacin’s MICs for most *A. baumannii* isolates (7/19) did not change more than fourfold in the presence of PAβN. This was also the case for the other antimicrobial agents that were assessed in the present study. Results from a study of 103 *Acinetobacter* isolates in Tehran showed a 40% lower MIC of ciprofloxacin when PaβN was added, but only 6.10% changed to susceptible MIC values [49]. Furthermore, Golanbar et al. also demonstrated a reduction in MIC when using PAβN [50], while Ribera et al. did not observe an effect on ciprofloxacin’s MIC when PaβN was added [47]. PAβN has generally and extensively been used as an efflux pump inhibitor in previous studies [49,51,52,53].

According to Coyne et al., a comparison of resistance levels in clinical *A. baumannii* MDR strains confirms that efflux is a major factor for resistance to various classes of drugs, including β-lactams, chloramphenicol, macrolides, tetracyclines, and aminoglycosides, with a high-level resistance to fluoroquinolones that requires additional mechanisms, such as the alteration of DNA type II isomerases [27]. Cheng et al. found that the use of fluoroquinolones predisposes a high colonization density of MDR in nasal and fecal specimens [54]. Efflux pumps have an important role in developing *Acinetobacter* spp. antimicrobial resistance, along with an overexpression of AmpC, β-lactamases, or carbapenemases [13]. This was also observed in the analyzed strains, as they expressed oxacillinases (OXA-23, OXA-24, and OXA-143) along with efflux pumps in a previous study [55].

It has been described that *A. baumannii*’s resistance to aminoglycosides results from the production of aminoglycoside-modifying enzymes (AMEs). Their action is selective, so they affect various aminoglycoside molecules differently. Although AMEs are considered the main mechanism of resistance to aminoglycosides in *A. baumannii*, these antibiotics are also subject to the action of efflux pumps, although they are described as less efficient in the extrusion of amikacin [56,57,58]. However, our results describe a significant increase in sensitivity in the presence of the inhibitor plus amikacin.

A paradoxical effect was observed in some isolates that were subjected to ciprofloxacin and levofloxacin and showed an increased resistance in the presence of the inhibitor. This effect has been described for both the quinolone family and the PAβN inhibitor [59,60]. Known as the Eagle effect or paradoxical growth, it is a phenomenon in which bacteria exposed to antibiotic concentrations that are higher than the optimal bactericidal concentration (OBC) paradoxically increase their survival levels due to a lower net rate of cell death. In this context, despite extensive observational reports on this effect in different microorganisms, its underlying mode of action is still not understood [60].

It has been widely described and demonstrated in the specialized medical literature that efflux pump inhibitors have enormous potential for use in combination therapy by increasing the efficacy of existing antimicrobials against multi-resistant pathogens such as *A. baumannii*. However, as can be seen in the results of the present study, which evaluated the sensitivity of clinical strains against an antibiotic together with a general efflux pump inhibitor, the main challenge is to recognize that the pathogen can express multiple efflux pumps with overlapping substrate profiles. It is for this reason that the search for a specific inhibitor of a particular pump is not very feasible due to the very nature of the expression of bacterial proteins, which allows for the expression of other simultaneous active pumps in the same strain. In this context, confirming the specificity of the inhibitor would require the silencing or elimination of other expressed pumps with simultaneous overlapping substrate profiles. Added to this is the fact that RND efflux pumps are under very strict regulatory control; not all strains express the desired pump to be examined in inhibition studies, which represents a limitation of these studies [41]. Despite the existence of these limitations, in controlled laboratory situations, it was possible to demonstrate that efflux pump inhibitors such as PAβN are active against the AdeFGH pump of *A. baumannii*. This efflux pump inhibitor analysis is given on clinical isolates of *A. baumannii* and underscores the importance of silencing or eliminating other efflux pumps with overlapping substrate profiles to characterize the inhibitor. At the same time, it highlights the importance of the search for specific inhibitors of the RND pumps in *A. baumannii*, although the clinical utility of this requires complementary studies [41].

Although the efflux pump inhibitor does not show intrinsic antimicrobial activity, it can potentiate the activity of the drugs evaluated against clinical strains of *A. baumannii*, which shows that it could be useful as an adjuvant to some antibiotics in the treatment of infections caused by multiresistant *A. baumannii*. This was demonstrated in a study that used norfloxacin [61] and that presented results like those obtained in the present work with the quinolones and aminoglycosides evaluated.

The results of this study indicate that efflux pumps have a role in conferring resistance to *A. baumannii* clinical isolates. Thus, efforts should be aimed at effectively detecting this pathogen and its resistance mechanisms to improve healthcare standards and guide antimicrobial therapy. Moreover, further research is needed to find more suitable, new compounds, as well as effective therapies [15], against *A. baumannii*. Updated available data particularly support the development of efflux pump inhibitors to be used in combination with antibiotics [33]. Moreover, further studies are required to assess the efficacy and safety of such compounds alongside current therapy to decrease the burden of disease that is attributable to this successful pathogen.

## 4. Materials and Methods

### 4.1. Samples

This study included nineteen non-duplicate clinical samples that were identified as *Acinetobacter baumannii* by PCR and collected from inpatients at the National Institute of Neoplastic Diseases (INEN, Instituto Nacional de Enfermedades Neoplásicas) over a 24-month period (January 2014–December 2016) in Lima, Peru.

Isolates were obtained from blood, bronchial aspirate, soft tissues, cerebrospinal fluid, and urine. Samples were collected as part of the hospital’s infection surveillance and control program. They were then stored and frozen at −80 °C.

Strains were recovered as part of an outbreak study [55], and selected strains were included and further assessed by microbiological and molecular techniques at the biomedical laboratory, Universidad Peruana de Ciencias Aplicadas (UPC), and Instituto de Investigación Nutricional (IIN).

### 4.2. Bacterial Culture Conditions and Identification

The different clinical samples collected were cultured in petri dishes with tryptic soy agar (TSA), which is a general nutrient medium for the growth and isolation of aerobic and anaerobic bacteria, and incubated at 37 °C for approximately 24 h under aerobic conditions [62]. Molecular identification was carried out by amplification of the blaOXA-51-like gene by real-time polymerase chain reaction (real-time PCR). Before carrying out the amplification of the DNA fragment, the extraction of the bacterial genetic material was carried out using the protocol described by Oh et al. [63]. Then, for DNA amplification, previously described primers and probes directed to the blaOXA-51-like gene were used [64,65]. A commercial strain of *A. baumannii* (ATCC 19606) was used as a positive control. Amplified products were gel-recovered, purified (SpinPrepTM Gel DNA Kit, San Diego, CA, USA), and sent to be sequenced (Macrogen, Seoul, Korea).

The LightCycler 2.0 (Roche Diagnostic, Basel, Switzerland) was used for the detection and amplification of genetic material by real-time PCR. Conditions were initial 10 min at 95 °C, followed by 55 cycles of denaturation at 95 °C for 5 s, annealing at 60 °C for 5 s, and elongation at 72 °C for 15 s. Then, the melting curve protocol was applied under the conditions of 95 °C for 20 s and then increments of 0.2 °C/s executed between 40 °C and 85 °C. Data acquisition was obtained during hybridization.

All the bacteria isolated and included in the study are disposable for scientific non-commercial purposes.

### 4.3. Antimicrobial Susceptibility Testing

Antimicrobial susceptibility to quinolones (levofloxacin, ciprofloxacin) and aminoglycosides (tobramycin, amikacin) was assessed by broth microdilution technique according to the 2018 Clinical and Laboratory Standards Institute (CLSI) guidelines [35] and as Gholami et al. described for nalidixic acid [33,66]. The susceptibility test for the antimicrobials studied was performed using the Kirby–Bauer disk diffusion method. A bacterial suspension was obtained from cultures after evidencing growth. The turbidity of each bacterial suspension was adjusted to a value of 0.5 McFarland standard and then inoculated onto Müller–Hinton agar (Merck, Darmstadt, Germany). MIC concentrations were evaluated from 0.25 µg/mL up to 256 µg/mL in a 96-well microtiter plate with 100 µL of the antibiotic dilution and Müller–Hinton broth. The correct density of samples was standardized at 625 nm spectrophotometry, and samples were then incubated at 37 °C for 24 h. The MIC was the lowest concentration without detection of bacterial growth [33,35], and *Escherichia coli* ATCC 25922 was used as a quality control strain. Furthermore, information from the 2020 CLSI M100 guidelines [28] was used to determine the cut-off values, between sensitive, intermediate, and resistant, that were taken into consideration for the antimicrobials used in this study.

### 4.4. Phenylalanine–Arginine beta-Naphthylamide (PAβN)

The activity of the efflux pumps system was assessed through the addition of phenylalanine–arginine β-naphthylamide (inhibitor to the broth microdilution, as previously assessed in Gholami et al.’s study) according to the 2018 Clinical and Laboratory Standards Institute (CLSI) guidelines [33,35]. Likewise, the susceptibility to antibiotics was assessed in the presence and absence of the PAβN inhibitor (Sigma-Aldrich, St. Louis, MO, USA). The minimum inhibitory concentration for each antibiotic was calculated in the presence and absence of the PAβN inhibitor. A 4-fold or greater reduction in MIC values after inhibitor addition was considered as a criterion of significance [31,48].

### 4.5. Ethics Statement

The study protocol was approved by the Research Ethics Board of the Instituto de Investigación Nutricional (IIN). The samples of isolates were obtained at the clinical laboratory within the context of an infection surveillance that the Committee for the Control and Prevention of Intrahospital Infections of the National Institute of Neoplastic Diseases regulated in accordance with the Technical Standard No. 753-2004/Ministry of Health of Peru, as well as the International Ethical Guidelines for Health-Related Research Involving Humans. Hospital-acquired infections (HAIs), also called health-care-associated infections (HCAIs), are a public health problem and, under these provisions, the collection of samples was exempted from informed consent. Patient information was coded when collected to ensure anonymity and confidentiality, and their characteristics were evaluated from their clinical records.

## 5. Conclusions

Our results demonstrate that the use of PAβN can improve the in vitro susceptibility of *A. baumannii* against the antibiotics that were evaluated. The study is relevant as it provides evidence that this efflux pump inhibitor can be used in combination with antibiotic therapy. Efflux pumps are still an important mechanism of antimicrobial resistance in *Acinetobacter baumannii* strains, and the focus on the inhibitory effects of some substances, including PAβN, could provide insight as a complementary therapy or new antimicrobials in future research.

## Figures and Tables

**Figure 1 antibiotics-12-01071-f001:**
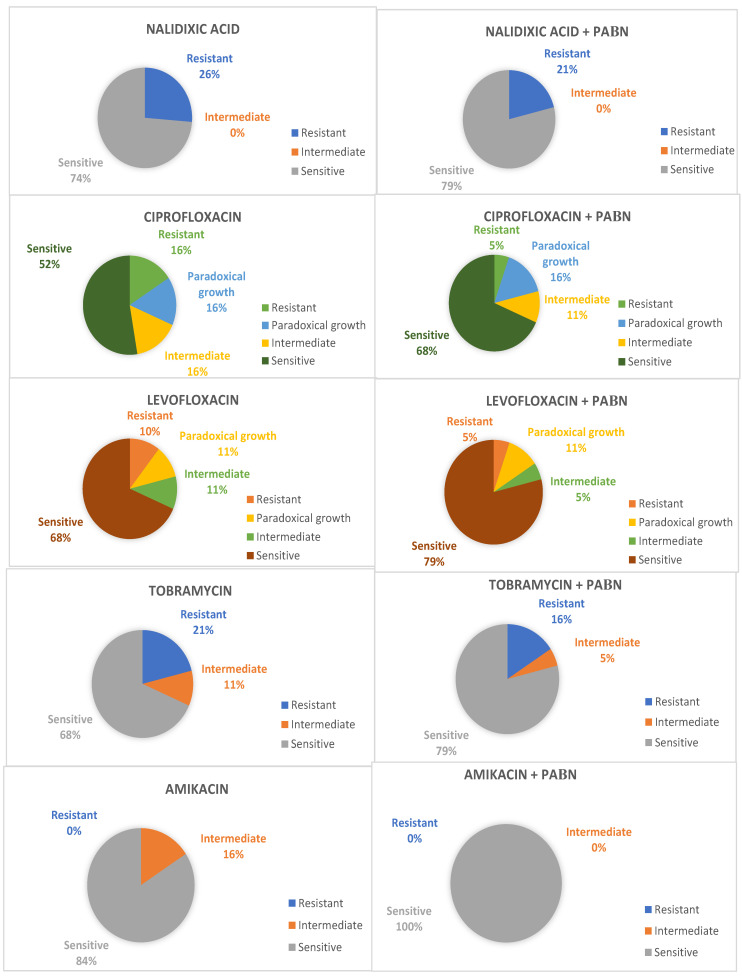
Change in susceptibility interpretation according to CLSI 2018 MIC values.

**Table 1 antibiotics-12-01071-t001:** Susceptibility profiles of the *Acinetobacter baumannii* strains included in this study ^a^.

Strain	MIC (µg/mL) (Susceptibility Rating) for Indicated Drug									
	Antibiotic					Antibiotic + PaβN				
	TOB	CIP	LEV	NA	AK	TOB	CIP	LEV	NA	AK
Ac2	0.5 (S)	0.5 (S)	8 (R)	128 (R)	<0.25 (S)	1 (S)	0.25 (S)	**2 (S) ^i^**	**32 (R)**	2 (S)
Ac10	1 (S)	0.5 (S)	32 (R)	32 (R)	<0.25 (S)	2 (S)	1 (S)	32 (R)	32 (R)	1 (S)
Ac20	8 (I)	2 (I)	1 (S)	1 (S)	<0.25 (S)	4 (S)	**0.5 (S) ^i^**	4 (I)	4 (S)	<0.25 (S)
Ac21	8 (I)	<0.25 (S)	<0.25 (S)	1 (S)	8 (S)	4 (S)	32 (R)	0.5 (S)	2 (S)	4 (S)
Ac22	0.5 (S)	0.25 (S)	2 (S)	16 (S)	4 (S)	0.25 (S)	0.5 (S)	4 (I)	**1 (S)**	2 (S)
Ac23	64 (R)	8 (R)	0.25 (S)	0.5 (S)	16 (S)	64 (R)	**2 (I) ^i^**	0.25 (S)	2 (S)	**4 (S)**
Ac24	0.5 (S)	2 (I)	0.5 (S)	1 (S)	32 (I)	1 (S)	**0.25 (S) ^i^**	0.25 (S)	0.5 (S)	16 (S)
Ac25	32 (R)	0.5 (S)	1 (S)	0.25 (S)	32 (I)	16 (R)	0.25 (S)	**0.25 (S)**	0.5 (S)	**4 (S) ^i^**
Ac26	0.25 (S)	8 (R)	0.5 (S)	64 (R)	<0.25 (S)	2 (S)	16 (R)	1 (S)	32 (R)	<0.25 (S)
Ac27	0.5 (S)	1 (S)	0.25 (S)	64 (R)	16 (S)	4 (S)	2 (I)	0.25 (S)	**8 (S)^i^**	**4 (S)**
Ac28	0.25 (S)	4 (R)	2 (S)	16 (S)	<0.25 (S)	4 (S)	**<0.25 (S) ^i^**	**0.5 (S)**	16 (S)	1 (S)
Ac29	0.5 (S)	2 (I)	1 (S)	64 (R)	32 (I)	4 (S)	2 (I)	1 (S)	32 (R)	**4 (S) ^i^**
Ac30	2 (S)	0.5 (S)	1 (S)	2 (S)	0.5 (S)	**0.5 (S)**	0.25 (S)	**<0.25 (S)**	**0.5 (S)**	0.25 (S)
Ac39	<0.25 (S)	<0.25 (S)	2 (S)	2 (S)	1 (S)	<0.25 (S)	<0.25 (S)	**0.5 (S)**	**0.25 (S)**	0.5 (S)
Ac41	16 (R)	<0.25 (S)	1 (S)	4 (S)	2 (S)	16 (R)	<0.25 (S)	2 (S)	2 (S)	**<0.25 (S)**
Ac46	16 (R)	<0.25 (S)	0.25 (S)	4 (S)	<0.25 (S)	8 (I)	0.25 (S)	<0.25 (S)	**1 (S)**	0.25 (S)
Ac50	1 (S)	<0.25 (S)	<0.25 (S)	0.25 (S)	16 (S)	0.5 (S)	4 (R)	<0.25 (S)	0.25 (S)	**<0.25 (S)**
Ac54	4 (S)	1 (S)	4 (I)	2 (S)	<0.25 (S)	**1 (S)**	1 (S)	2 (S)	**0.25 (S)**	4 (S)
Ac55	4 (S)	0.5 (S)	4 (I)	2 (S)	1 (S)	4 (S)	0.5 (S)	4 (I)	2 (S)	**<0.25 (S)**

^a^ TOB, tobramycin; CIP, ciprofloxacin; LEV, levofloxacin; NA, nalidixic acid; AK, amikacin; R, resistant; S, susceptible; I, intermediate. ^i^ Change in susceptibility interpretation according to CLSI 2018 MIC values. Bold: Fourfold change in MIC.

**Table 2 antibiotics-12-01071-t002:** Reference values of the susceptibility ranges of the antibiotics used according to the CLSI 2018 criteria.

Antimicrobial Agent	Disk Content	Interpretive Categories and MIC Breakpoints (µg/mL)		
		S	I	R
Tobramycin	10 µg	≤4	8	≥16
Amikacin	30 µg	≤16	32	≥64
Ciprofloxacin	5 µg	≤1	2	≥4
Levofloxacin	5 µg	≤2	4	≥8
Nalidixic acid	30 µg	≤16	-	≥32

R = resistant; S = susceptible; I = intermediate.

## Data Availability

Abstraction format used in the study and dataset are available and accessible from the corresponding author upon request at the link: https://figshare.com/s/d1cddba64d9769ffe92b. accessed on 16 January 2023.

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
