# Peer review of "Effect of Phenylalanine–Arginine Beta-Naphthylamide on the Values of Minimum Inhibitory Concentration of Quinolones and Aminoglycosides in Clinical Isolates of Acinetobacter baumannii"

_antibiotics, 2023, doi:10.3390/antibiotics12061071_

Round 1
Reviewer 1 Report
1. Why did the authors use the blaOXA-51-like gene to identify A. baumannii?
2. Provide the range of S, I, and R of antimicrobial drugs used in this study as recommended by CLSI; that would make it easier for readers to interpret the results.
3. Is it possible to include more tested isolates in this study? The nineteen isolates are probably insufficient to conclude that the PAβN really affects MIC values. Moreover, the results in Table 1 showed that some strains (such as Ac22, Ac30, Ac39, and Ac50) have no resistance to all antibiotics. Therefore, those might be sensitive strains that could be easily inhibited or killed by several kinds of inhibitors, which might not be good representatives for PAβN effect evaluation.
4. For the paradoxical effect
- Could you provide the list of isolates that showed paradoxical growth against fluoroquinolone antibiotics?
- Are there any other explanations for the situations in which the paradox of fluoroquinolone antibiotics may be relevant?
- What could be the reason why this phenomenon could not be observed by using other classes of antibiotic drugs?
5. In order to complete the efflux pump inhibitor study, could you determine the expression of RND regulating genes (or other efflux pump associated genes in A. baumannii) in both PAβN presence/absence groups to ensure that PAβN could reduce efflux pump capability in A. baumannii.
Author Response
Reviewer 01:
- Why did the authors use the blaOXA-51-likegene to identify baumannii?
Response: The authors understand the reviewer's question. The authors chose the blaOXA-51 like gene to identify A. baumannii for the following reasons:
- The blaOXA-51-like gene is used as a tool in molecular epidemiology studies for the characterization of Acinetobacter strains.
Rafei, R., Pailhoriès, H., Hamze, M., Eveillard, M., Mallat, H., Dabboussi, F., Joly-Guillou, M. L., & Kempf, M. (2015). Molecular epidemiology of Acinetobacter baumannii in different hospitals in Tripoli, Lebanon using bla(OXA-51-like) sequence based typing. BMC microbiology, 15, 103.
- We know that there is a variety of genes or specific DNA sequences of A. baumannii that can be used for identification by various PCR methods. The blaOXA-51 like gene was chosen based on:
The exact sequence of the target region is known, identifying itself as specific for A baumannii.
Guclu, A.U., & Gozen, A.G. (2020). Genetic Diversity of OXA-like Genes in Multidrug-Resistant Acinetobacter baumannii Strains from ICUs. Clinical laboratory, 66(10), 10.7754/Clin.Lab.2020.200135.
The primer has no homology to other unwanted sequences or samples. Non-specific amplifications and erroneous results have not been described.
Primers have been validated prior to use.
Turton, J.F., Woodford, N., Glover, J., Yarde, S., Kaufmann, M.E., & Pitt, T.L. (2006). Identification of Acinetobacter baumannii by detection of the blaOXA-51-like carbapenemase gene intrinsic to this species. Journal of clinical microbiology, 44(8), 2974–2976.
Ghaith, D.M., Hassan, R.M., & Hasanin, A.M. (2015). Rapid identification of nosocomial Acinetobacter baumannii isolated from a surgical intensive care unit in Egypt. Annals of Saudi medicine, 35(6), 440–444.
Gözalan A, AydoÄŸan S, HacıseyitoÄŸlu D, Kuzucu Ç, Köksal F, Açıkgöz ZC, Durmaz R. Acinetobacter baumannii Kan İzolatlarının MALDI-TOF MS, ARDRA ve blaOXA-51-benzeri Gene Özgül Gerçek Zamanlı Polimeraz Zincir Reaksi yonu ile Tanımlanması [The Identification of Acinetobacter baumannii Blood Isolates by MALDI-TOF MS, ARDRA and blaOXA-51-like Gene-Specific Real-Time PCR]. Mikrobiyol Bul. 2020 Oct;54(4):535-546. Turkish.
Vijayakumar S, Biswas I, Veeraraghavan B. Accurate identification of clinically important Acinetobacter spp.: an update. Future SciOA. 2019 Jun 27;5(6):FSO395.
- Provide the range of S, I, and R of antimicrobial drugs used in this study as recommended by CLSI; that would make it easier for readers to interpret the results.
Response: The authors agree with the reviewer's suggestion. The range of S, I and R is provided for each of the drugs used in the present study. Table 2 is presented for greater clarity in the information.
- Is it possible to include more tested isolates in this study? The nineteen isolates are probably insufficient to conclude that the PAβN really affects MIC values. Moreover, the results in Table 1 showed that some strains (such as Ac22, Ac30, Ac39, and Ac50) have no resistance to all antibiotics. Therefore, those might be sensitive strains that could be easily inhibited or killed by several kinds of inhibitors, which might not be good representatives for PAβN effect evaluation.
Response: The authors understand the reviewer's concern about the number of samples. However, we consider that the number of isolates obtained is sufficient due to the nature of these studies that present logistical challenges in terms of obtaining clinical samples to be analyzed and the budgetary limitations involved in analyzing a large volume of samples with pump inhibitors. of efflux. In this context, we are seeking funding to develop a multicenter study, with a greater number of clinical isolates that can broaden our knowledge of the subject.
Regarding strains AC22, AC30, AC39 and AC50, the authors understand the comment of the reviewer. However, we must highlight the usefulness of our analysis and results, not only based on categorizing what was obtained as sensitive or resistant, but also focusing on the value of that sensitivity and whether it can be improved by the action of the efflux pump inhibitor. As we mentioned in the results section and then in the discussion section, we not only believe, but we are sure that it is very useful for an antimicrobial to go from 0.5 µg /ml to 0.25 µg /ml in the presence of the inhibitor within the sensitive spectrum. This effect can also be explained as an effect of the inhibitor and contributes to its understanding.
Due to the limitations stated in previous lines, we do not consider it feasible to add more samples to the present study.
- For the paradoxical effect
- Could you provide the list of isolates that showed paradoxical growth against fluoroquinolone antibiotics?
Response: The authors agree with the reviewer's question. We indicate that the samples that registered a paradoxical growth were:
In the case of Ciprofloxacin: Ac21, Ac22, Ac27, Ac50
In the case of Levofloxacin: Ac22
For fluoroquinolones, 4 isolates presented paradoxical growth, one of them (Ac22), both against Ciprofloxacin and Levofloxacin.
- Are there any other explanations for the situations in which the paradox of fluoroquinolone antibiotics may be relevant?
Response: The authors consider the reviewer's question very pertinent. In this regard, in relation to other relevant explanations, they indicate that bacteria can acquire mutations in the topoisomerase enzyme (main objective of quinolones) which allows them to resist the effects of antibiotics. In the presence of quinolones, these resistant bacteria may have a selective advantage over sensitive bacteria, resulting in faster and seemingly paradoxical growth. In this context, it is important to consider that paradoxical bacterial growth with quinolones does not occur in all bacterial species or in all cases. Understanding the mechanisms behind paradoxical bacterial growth is still under investigation and further studies are needed to get a full picture of this phenomenon.
- What could be the reason why this phenomenon could not be observed by using other classes of antibiotic drugs?
Response: The authors agree to deepen the reviewer's question. The understanding of the mechanisms behind the paradoxical bacterial growth is currently the subject of research in the world to achieve its understanding. However, a possible answer lies in the mechanism of action of quinolones, which may have divergent effects on bacterial gene expression depending on the concentration of the antibiotic. At low concentrations, bacterial DNA replication is inhibited, which stops bacterial growth. At high concentrations, a stress response can be induced in bacteria, which can activate repair systems and increase the synthesis of protective enzymes. This can allow the bacteria to survive and grow even in the presence of the antibiotic. This mechanism of action belongs to the quinolones, so it does not apply to other classes or families of antibiotics. Some authors propose the existence of an antibiotic pressure threshold.
Carret G, Flandrois JP, Lobry JR. Biphasic kinetics of bacterial killing by quinolones. J Antimicrob Chemother. 1991 Mar;27(3):319-27
García-Rey C, Martín-Herrero JE, Baquero F. Antibiotic consumption and generation of resistance in Streptococcus pneumoniae: the paradoxical impact of quinolones in a complex selective landscape. Clin Microbiol Infect. 2006 May;12 Suppl 3:55-66.
Smirnova GV, Oktyabrsky ON. Relationship between Escherichia coli growth rate and bacterial susceptibility to ciprofloxacin. FEMS Microbiol Lett. 2018 Jan 1;365(1)
- In order to complete the efflux pump inhibitor study, could you determine the expression of RND regulating genes (or other efflux pump associated genes in baumannii) in both PAβN presence/absence groups to ensure that PAβN could reduce efflux pump capability in A. baumannii.
Response: The authors understand the reviewer's suggestion. A limitation for the present study is not being able to continue or add to determine the expression of RND regulatory genes. We have two main reasons for this:
- The analysis of the expression of regulatory genes of RND and other genes associated with efflux pumps in A. baumannii is challenging due to the complexity of the system. RNDs are multidrug efflux transport systems. These systems are regulated by multiple genes and signaling pathways.
- Technical and cost limitations: Determining the expression of RND regulatory genes and other genes associated with expulsion pumps exceeded the original project budget.

Reviewer 2 Report
In this Manuscript Rebata et al. provide the important findings such as identification of MIC using efflux pump inhibitor drugs in combination with antibiotics to treat the clinical isolates- Acinobacter strains .
This approach of identifying efflux pump inhibitors (adjuvant drugs) and treating in combination with antibiotic drugs would provide a better therapeutic approach and possible hope in the successive clinical trials.
The manuscript was well written, the experiments well designed, and the conclusions are appropriate. I believe that the findings of the manuscript are of sufficient novelty and breadth to merit publication in antibiotics journal.
Below are suggestions for minor revision of the manuscript:
1. The authors need to check whether adjuvant drug by itself has any activity on the clinical isolates.
2. sentences in abstract authors ended bluntly.It would be better to revise abstract part in the article.
3.Line 22 in abstract bacteria "expresses several resistance mechanisms" would be better to include appropriate word.
4. Methods section: instead of saying Efflux pump inhibitor the alternate heading would be better to include.
I have no major concerns with this paper to publish in Antibiotics Journal with minor revision.
The abstract portion of the article writing would have been better. However introduction and discussion part was written well.
Author Response
Reviewer 02:
- The authors need to check whether adjuvant drug by itself has any activity on the clinical isolates.
Response: The authors consider the reviewer's question very pertinent. We have carried out a rigorous search of the information and based on this we verified the following:
The mechanism of action of PAβN lies in the inhibition of the RND efflux pumps, which prevents the expulsion of antibiotics and other compounds. By doing so, PAβN increases the susceptibility of bacteria to antibiotics and can restore their efficacy. On this basis PAβN itself does not have intrinsic antibiotic properties.
Mahamoud, A., Chevalier, J., Alibert-Franco, S., Kern, W.V., & Pagès, J.M. (2007). Antibiotic efflux pumps in Gram-negative bacteria: the inhibitor response strategy. Journal of Antimicrobial Chemotherapy, 59(6), 1223-1229.
Lamers, R.P., Cavallari, J.F., & Burrows, L.L. (2013). The efflux inhibitor phenylalanine-arginine beta-naphthylamide (PAβN) permeabilizes the outer membrane of gram-negative bacteria. PloS one, 8(3), e60666.
- sentences in abstract authors ended bluntly.It would be better to revise abstract part in the article.
Response: The authors agree with the reviewer's suggestion. The summary is reviewed and corrected for better understanding.
- Line 22 in abstract bacteria "expresses several resistance mechanisms" would be better to include appropriate word.
Response: The authors agree with the reviewer's suggestion. The sentence is restructured for better understanding.
- Methods section: instead of saying Efflux pump inhibitor the alternate heading would be better to include.
Response: The authors agree with the reviewer's suggestion. The name of the inhibitor studied is placed in the methods section.

Round 2
Reviewer 1 Report
The authors have addressed all questions.
This manuscript can be published in this journal in current form.